# Chlorophyll Fluorescence Imaging Analysis for Elucidating the Mechanism of Photosystem II Acclimation to Cadmium Exposure in the Hyperaccumulating Plant *Noccaea caerulescens*

**DOI:** 10.3390/ma11122580

**Published:** 2018-12-18

**Authors:** Gülriz Bayçu, Julietta Moustaka, Nurbir Gevrek, Michael Moustakas

**Affiliations:** 1Division of Botany, Department of Biology, Faculty of Science, Istanbul University, 34134 Istanbul, Turkey; gulrizb@istanbul.edu.tr (G.B.); ngevrek@gmail.com (N.G.); 2Department of Plant and Environmental Sciences, University of Copenhagen, Thorvaldsensvej 40, DK-1871 Frederiksberg C, Denmark; moustaka@plen.ku.dk; 3Department of Botany, Aristotle University of Thessaloniki, 54124 Thessaloniki, Greece

**Keywords:** Cd toxicity, detoxification mechanism, photochemical quenching, photosynthetic heterogeneity, photoprotective mechanism, phytoremediation, plastoquinone pool, redox state, spatiotemporal variation

## Abstract

We provide new data on the mechanism of *Noccaea caerulescens* acclimation to Cd exposure by elucidating the process of photosystem II (PSII) acclimation by chlorophyll fluorescence imaging analysis. Seeds from the metallophyte *N. caerulescens* were grown in hydroponic culture for 12 weeks before exposure to 40 and 120 μM Cd for 3 and 4 days. At the beginning of exposure to 40 μM Cd, we observed a spatial leaf heterogeneity of decreased PSII photochemistry, that later recovered completely. This acclimation was achieved possibly through the reduced plastoquinone (PQ) pool signaling. Exposure to 120 μM Cd under the growth light did not affect PSII photochemistry, while under high light due to a photoprotective mechanism (regulated heat dissipation for protection) that down-regulated PSII quantum yield, the quantum yield of non-regulated energy loss in PSII (Φ*_NO_*) decreased even more than control values. Thus, *N. caerulescens* plants exposed to 120 μM Cd for 4 days exhibited lower reactive oxygen species (ROS) production as singlet oxygen (^1^O_2_). The response of *N. caerulescens* to Cd exposure fits the ‘Threshold for Tolerance Model’, with a lag time of 4 d and a threshold concentration of 40 μM Cd required for the induction of the acclimation mechanism.

## 1. Introduction

Cadmium is a non-essential heavy metal that can occur in the environment in high concentrations as a consequence of numerous human activities, thus becoming toxic to all organisms [1,2,3,4,5]. Plants have developed several exclusive and effective mechanisms for Cd detoxification and tolerance, including control of Cd influx and acceleration of Cd efflux, Cd chelation and sequestration, Cd remobilization, and scavenging of Cd-induced reactive oxygen species [5,6,7,8,9,10].

Hyperaccumulators are plant species that vigorously take up heavy metals, translocate them into the above-ground parts and isolate them into a risk-free state [4,11]. These plants can accumulate several percent of heavy metals in their dry mass [4]. Hyperaccumulators also have to stock the absorbed heavy metal in a manner that is not detrimental to vital enzymes and especially photosynthesis [12,13]. Hyperaccumulators can be used for phytoremediation and also for phytomining [4,14,15,16]. Phytoremediation is a cost-effective and environmentally-friendly technology that uses plants to remove the toxic metals from soils; it has been widely used in practice [14,17].

*Noccaea caerulescens* is known as a zinc–cadmium–nickel hyperaccumulator because it can accumulate these metals at extremely high concentrations in its aboveground tissues [18], and has been proposed as an ideal species for examining metal tolerance and hyperaccumulation [19]. It has recently gained a lot of attention due to its potential use in phytoremediation and phytomining [20,21]. Certain ecotypes of *N. caerulescens* can store as much as 14,000 μg Cd g^−1^ dry biomass without showing toxicity signs [22,23,24]. Cadmium concentrations in the leaves above 0.01% dry biomass are considered extraordinary and are the limit level for Cd hyperaccumulation [24,25].

Photosynthesis has been shown to be very sensitive to Cd either directly or indirectly [4,26,27,28,29,30,31]. A Cd-induced decrease in photosynthetic efficiency may result from disturbances in the electron transport [32,33], enzymatic activities involved in CO_2_ fixation [34,35], or from stomatal closure [36,37]. Photosystem II (PSII) is extremely sensitive to Cd that exerts multiple effects on both donor (it inhibits oxygen evolution) and acceptor sites (it inhibits electron transfer from quinone A, QA to quinone B, QB) [28,33,38,39]. The less susceptible component of the photosynthetic apparatus to Cd is thought to be PSI [28,40].

We investigated photosynthetic acclimation to Cd toxicity using the hyperaccumulator *Noccaea caerulescens*. Our previous study indicated that despite the substantial high toxicity levels of Zn and Cd in *N. caerulescens* aboveground tissues, the photochemical energy use at PSII did not differ compared to controls [13]. However, the underlying mechanism of photosynthetic acclimation has not been elucidated. In the present study, in order to investigate the mechanism of *N. caerulescens* acclimation to Cd exposure and to clarify the process of photosynthetic acclimation, we treated in hydroponic culture *N. caerulescens* plants with 40 and 120 μM Cd for 3 and 4 days.

## 2. Materials and Methods

### 2.1. Seed Collection and Experimental Design

Seeds of *Noccaea caerulescens* F.K. Mey collected from a former Copper Mine area at Røros (Norway) were cultivated hydroponically in an environmental growth chamber as described previously [13].

After growth for 12 weeks, some plants were exposed to 40 or 120 μM Cd (supplied as 3CdSO_4_.8H_2_O) for 3 and 4 days while others were left to control growth conditions.

### 2.2. Chlorophyll Fluorescence Imaging Analysis

Chlorophyll fluorescence measurements were carried out with an Imaging-PAM Chlorophyll Fluorometer (Walz, Effeltrich, Germany) in dark-adapted leaves (15 min) of *N. caerulescens* plants, grown at 0 (control), 40 or 120 μM Cd for 3 and 4 days, as described previously [13,41]. Five leaves were measured from five different plants with eight areas of interest in each leaf. Two light intensities were selected for chlorophyll fluorescence measurements, a low light intensity that was similar to the growth light (300 μmol photons m^−2^ s^−1^, GL) and a high light intensity (1000 μmol photons m^−2^ s^−1^, HL, more than three times that of the growth light). The measured and calculated chlorophyll fluorescence parameters with their definitions are given in Table 1.

Representative results of the measured chlorophyll fluorescence parameters are also displayed as color-coded images.

### 2.3. Statistical Analyses

All measurements that are expressed as mean ± SD were analyzed by student *t*-test (*p* < 0.05). Five leaves from five different plants were analyzed in each treatment. In all graphs, the error bars are standard deviations, while columns with the same letter are not statistically different at *p* < 0.05.

## 3. Results

### 3.1. Changes in the Maximum Quantum Efficiency of PSII Photochemistry after Cd Exposure

At the beginning of exposure to 40 μM Cd, the maximum quantum efficiency of PSII photochemistry (*F*_v_/*F*_m_) in *N. caerulescens* decreased significantly but increased to control values at 120 μM Cd (Figure 1).

### 3.2. Changes in the Allocation of Absorbed Light Energy in PSII after Cd Exposure

The quantum yield of photochemical energy conversion in PSII (Φ*_PSII_*), at both growth light (GL) and high light (HL) intensity decreased significantly compared to the control, after 3 d at 40 μM Cd, while it improved during the 4 d (Figure 2). However, Φ*_PSII_* increased to control values after 3 d at 120 μM Cd at GL and stabilized to control values after 4 days of exposure (Figure 2a). High light (HL) exposure to 120 μM Cd resulted in decreased Φ*_PSII_* compared to controls (Figure 2b).

The quantum yield of regulated non-photochemical energy loss in PSII (Φ*_NPQ_*) decreased significantly compared to the control after 3 d at 40 μM Cd at GL and increased to control values during the 4 d (Figure 3a). Exposure to 120 μM Cd resulted in decreased Φ*_NPQ_* at GL compared to controls during the 4 d (Figure 3a). At HL, Φ*_NPQ_* remained unchanged at 40 μM Cd, but increased significantly at 120 μM Cd (Figure 3b).

The quantum yield of non-regulated energy loss in PSII (Φ*_NO_*), a loss process due to PSII inactivity, at both GL and HL intensity, increased significantly compared to the control after 3 d exposure to 40 μM Cd, while during the 4 d it decreased compared to 3 d (Figure 4). After exposure to 120 μM Cd for 3 d at GL, Φ*_NO_* retained the same values compared to the controls, but increased during the 4 d (Figure 4a). However, Φ*_NO_* decreased more than the control values at 120 μM Cd at HL (Figure 4b).

### 3.3. Non-Photochemical Quenching and Electron Transport Rate in Response to Cd

Non-photochemical quenching (NPQ) that reflects heat dissipation of excitation energy, decreased significantly compared to the control after 3 d at 40 μM Cd at GL, while it improved during the 4 d (Figure 5a). Exposure to 120 μM Cd resulted in decreased NPQ at GL compared to controls during the 4 d (Figure 5a). At HL, NPQ decreased significantly compared to the control after 3 d exposure to 40 μM Cd, and increased to control values during the 4 d, while after exposure to 120 μM Cd increased significantly compared to the controls (Figure 5b).

The electron transport rate (ETR), at both GL and HL intensity, decreased significantly compared to the control after 3 d at 40 μM Cd, while it improved during the 4 d (Figure 6). However, ETR increased to control values after 3 d exposure to 120 μM Cd at GL and stabilized to control values after 4 d exposure (Figure 6a). High light exposure to 120 μM Cd resulted in decreased ETR compared to the controls (Figure 6b).

### 3.4. Changes in the Redox State of PSII after Cd Exposure

The redox state of QA (*q*_P_) that is a measure of the fraction of open PSII reaction centers, at both GL and HL intensity, decreased significantly compared to the control after 3 d at 40 μM Cd, while it improved during the 4 d (Figure 7). However, *q*_P_ increased to control values after 3 d exposure to 120 μM Cd at GL and stabilized to control values after 4 d exposure (Figure 7a). High light exposure to 120 μM Cd resulted in a more reduced redox state of QA compared to controls, i.e., a lower fraction of open PSII reaction centers (Figure 7b).

### 3.5. Spatiotemporal Variation of PSII Responses to Cd Exposure

The major veins (mid-vein, first- and second-order veins) in *N. caerulescens* leaves grown under control growth conditions at both GL and HL defined areas with a lower fraction of open PSII reaction centers or a more reduced redox state of QA, while mesophyll cells expressed larger spatial heterogeneity with a larger fraction of open PSII reaction centers or a more oxidized redox state (Figure 8e and Figure 9d).

The maximum quantum efficiency of PSII photochemistry (*F*_v_/*F*_m_) show the smallest spatial heterogeneity even though it decreased significantly at 40 μM Cd and increased to control values at 120 μM Cd (Figure 8a). The quantum yield of photochemical energy conversion in PSII (Φ*_PSII_*) decreased significantly after 3 d at 40 μM Cd at GL, while it improved during the 4 d, showing a high spatiotemporal leaf heterogeneity (Figure 8b). Among the chlorophyll fluorescence parameters with high spatiotemporal heterogeneity observed at GL, were the images of the quantum yield of non-regulated energy dissipated in PSII (non-regulated heat dissipation, a loss process due to PSII inactivity) (Φ*_NO_*) (Figure 8d) and the images of the redox state of the PQ pool (*q*_P_) (Figure 8e). The most severely affected leaf area after 3 d at 40 μM Cd, was the left and right leaf side, while the central area was less affected (Figure 8d,e). At the left and right leaf side after 3 d exposure to 40 μM Cd, the quantum yield of non-regulated energy loss in PSII (Φ*_NO_*) increased; thus, these areas exhibited increased singlet oxygen (^1^O_2_) production (Figure 8d), and also presented the lower *q*_P_ values (Figure 8e). However, in the left and right leaf side after 4 d exposure to Cd, Φ*_NO_* decreased (Figure 8d) and the redox state of the PQ pool increased (*q*_P_) (Figure 8e). At exposure to 120 μM Cd at GL, leaf spatial heterogeneity decreased, and both Φ*_NO_* (Figure 8d) and *q*_P_ (Figure 8e) stabilized to control values.

Exposure of *N. caerulescens* to HL increased the spatiotemporal leaf heterogeneity (Figure 9) and the plants suffered more from Cd toxicity during the 3 d of exposure to 40 μM Cd, but they recovered during the 4 d. However, exposure to 120 μM Cd at HL revealed mild effects. This was realized by an increase in Φ*_NPQ_* (Figure 9b) that down-regulated PSII quantum yield (Φ*_PSII_*) (Figure 9a) and decreased the quantum yield of non-regulated energy loss in PSII (Φ*_NO_*) (Figure 9c).

## 4. Discussion

The type of damage on PSII that has frequently been identified as the main target of Cd toxicity on photosynthesis strongly depends on light conditions [4,43,44,45,46]. At GL, the damage of the PSII function is mainly due to the impairment that results from the replacement by Cd^2+^ of the Mg^2+^ ion in the chlorophyll molecules of the light-harvesting complex II, while in HL it is mainly from direct damage to the PSII reaction center [4,44,45,46].

*N. caerulescens* leaves grown under control growth conditions at both GL and HL show a spatial heterogeneity in PSII functionality (Figure 8 and Figure 9). This spatial heterogeneity may be attributed to ‘patchy stomatal behavior’, in which stomata in adjacent regions exhibit significantly different mean apertures from each other, resulting in significantly different stomatal conductance (g_s_) [47,48]. Stomatal conductance decreases when the stomata close; this is used as an indicator of the extent of stomatal opening [49,50]. It is assumed that spatial variation in the quantum efficiency of PSII photochemistry (Φ*_PSII_*) arises from local differences in internal CO_2_ concentrations, which in turn result from changes in stomatal conductance due to patchy stomatal behavior [51]. A body of evidence suggests that patterns of Φ*_PSII_* can be used to calculate stomatal conductance [51,52,53,54,55].

At the beginning of exposure to 40 μM, Cd Φ*_PSII_* decreased significantly at the left and right leaf sides (Figure 8b and Figure 9a), with a simultaneous decrease in Φ*_NPQ_* (Figure 8c and Figure 9b) resulting in an increase of the quantum yield of non-regulated non-photochemical energy loss (Φ*_NO_*) (Figure 8d and Figure 9c). The increase in Φ*_NO_* indicates that photochemical energy conversion and photoprotective regulatory mechanism were insufficient, pointing to serious problems of the plant to cope with the absorbed light energy [56,57]. Φ*_NO_* consists of chlorophyll fluorescence internal conversions and intersystem crossing, which indicate the formation of singlet oxygen (^1^O_2_) via the triplet state of chlorophyll (^3^chl *) [13,58,59]. After 3 d exposure to 40 μM Cd, *N. caerulescens* leaves exhibited increased ^1^O_2_ production at the left and right leaf sides, since Φ*_NO_* increased significantly at those areas. Thus, although Cd^2+^ is a redox-inert element, it produces reactive oxygen species [28]. The simultaneous reduced PQ pool that was observed mainly at the left and right leaf sides mediated stomatal closure probably through the generation of mesophyll chloroplastic hydrogen peroxide (H_2_O_2_) [60]. The stomatal closure at these areas implies decreased transpiration rates that slow down Cd supply.

During the 4 d exposure to 40 μM Cd, Φ*_PSII_* increased at the left and right leaf sides (Figure 8b and Figure 9a), with a simultaneous increase in Φ*_NPQ_* (Figure 8c and Figure 9b) resulting in a decrease of Φ*_NO_* (Figure 8d and Figure 9c) compared to 3 d exposure. This response is attributed to both the possible Cd detoxification mechanism achieved by vacuolar sequestration, that seems to be the main mechanism for Cd detoxification [61,62,63], and to the reduced plastoquinone (PQ) pool that mediated stomatal closure and decreased Cd supply at the affected leaf area, leading to the acclimation of *N. caerulescens* to Cd exposure. Under exposure to 120 μM Cd at HL, the quantum yield of non-regulated energy loss in PSII (Φ*_NO_*) decreased even more than control values, and thus exhibited lower singlet oxygen (^1^O_2_) production. This was due to the photoprotective mechanism that can divert absorbed light to other processes such as thermal dissipation, preventing the photosynthetic apparatus from oxidative damage [64,65,66,67,68,69,70].

The observed spatial heterogeneity in the quantum yield of linear electron transport (Φ*_PSII_*) in *N. caerulescens* leaves exposed to 40 μM Cd for 3 d (Figure 8b and Figure 9a) is in accordance to elemental imaging using laser ablation inductively-coupled plasma mass spectrometry, performed on whole leaves of the hyperaccumulator *N. caerulescens* that revealed differences in the supply of Cd over the whole leaf area, suggesting a heterogeneous distribution across the leaf [71]. Useful information can be obtained by combining chlorophyll fluorescence images, followed by laser ablation inductively-coupled plasma mass spectrometry on whole leaves of the hyperaccumulator *N. caerulescens* exposed to Cd.

It seems that spatiotemporal variations in the redox state of the PQ pool related to stomatal conductance, an indicator of the extent of stomatal opening [50], are interconnected to the heterogeneous distribution of Cd over the entire leaf area [71]. Thus, the spatial heterogeneity in the redox state of the PQ pool throughout the whole leaf area (Figure 8e and Figure 9d) reveals a spatial supply of Cd across the leaf. Recently, Cd^2+^ root influx has been shown to exhibit spatiotemporal patterns [72]. A heterogeneous distribution of a reduced PQ pool gives rise to a spatial distribution of H_2_O_2_ accumulation [73]. Still, reactive oxygen species (O_2_^−^, H_2_O_2_) production corresponds to spatial accumulation metal patterns [74].

In our work, the response of *N. caerulescens* to Cd exposure fits the ‘Threshold for Tolerance Model’, with a lag time or/and a threshold concentration required for the induction of a tolerance mechanism [75,76,77,78]. Concurrent to this model, mild stress or short exposure times can produce significant effects on plants, while moderate stress or longer exposure times have less or no effect [79]. In accordance with this model, 40 μM Cd and 3d exposure time caused significant effects on PSII functioning, while 120 μM Cd or 4d exposure time have less or no effect. A lag-time of 4d exposure to 40 μM Cd was required for *N. caerulescens* to activate stress-coping mechanisms.

## 5. Conclusions

Acclimation to Cd exposure was achieved through the possible Cd detoxification mechanism done by vacuolar sequestration and the reduced plastoquinone (PQ) pool signaling that mediated stomatal closure and decreased Cd supply at the affected leaf area. The response of *N. caerulescens* to Cd exposure fits the ‘Threshold for Tolerance Model’, with a lag time of 4 d and a threshold concentration of 40 μM Cd required for the induction of the acclimation mechanism through the reduced PQ pool that mediated stomatal closure probably by the generation of mesophyll chloroplastic hydrogen peroxide (H_2_O_2_) [60], which acts as a fast acclimation signaling molecule [73,80], as well as activates the Cd detoxification mechanism through vacuolar sequestration [61,62,63]. The mode of Cd damage on PSII strongly depends on the irradiance conditions [4,43,44,45,46]. Chlorophyll fluorescence imaging analysis is a non-invasive tool to assess the physiological status of plants and detect the impacts of environmental stress [81,82,83], permitting also the visualization of the spatiotemporal variations in PSII efficiency [76]. As it was shown in our experiments, it is also capable of elucidating the mechanism of photosystem II acclimation to Cd exposure.

## Figures and Tables

**Figure 1 materials-11-02580-f001:**
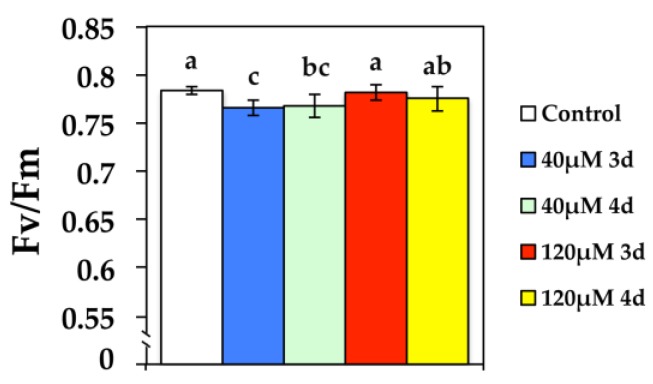
Changes in the maximum quantum efficiency of PSII (*F*_v_/*F*_m_) in *N. caerulescens* plants grown at 0 (control), 40 or 120 μM Cd^2+^ for 3 and 4 days.

**Figure 2 materials-11-02580-f002:**
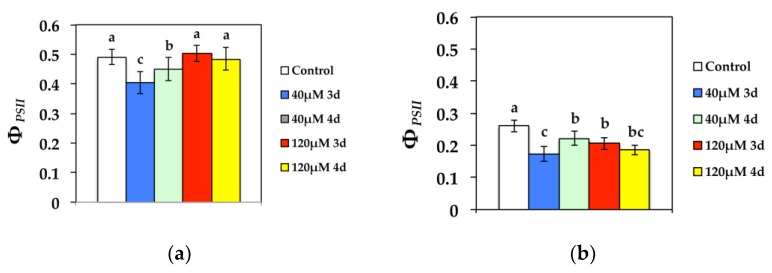
Changes in the quantum efficiency of PSII photochemistry (Φ*_PSII_*) in *N. caerulescens* measured (**a**) at 300 μmol photons m^−2^ s^−1^ or (**b**) 1000 μmol photons m^−2^ s^−1^. *N. caerulescens* plants were grown at 0 (control), 40, or 120 μM Cd^2+^ for 3 and 4 days.

**Figure 3 materials-11-02580-f003:**
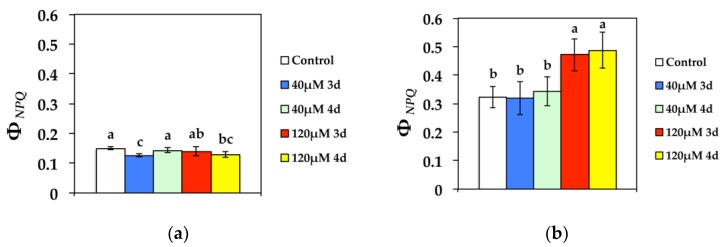
Changes in the quantum yield for dissipation by down regulation in PSII (regulated heat dissipation, a loss process serving for protection) (Φ*_NPQ_*) measured at (**a**) 300 μmol photons m^−2^ s^−1^ or (**b**) 1000 μmol photons m^−2^ s^−1^. *N. caerulescens* plants were grown at 0 (control), 40, or 120 μM Cd^2+^ for 3 and 4 days.

**Figure 4 materials-11-02580-f004:**
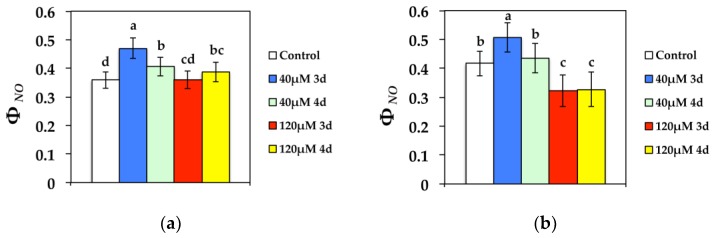
Changes in the quantum yield of non-regulated energy dissipated in PSII (non-regulated heat dissipation, a loss process due to PSII inactivity) (Φ*_NO_*) measured at (**a**) 300 μmol photons m^−2^ s^−1^ or (**b**) 1000 μmol photons m^−2^ s^−1^. *N. caerulescens* plants were grown at 0 (control), 40, or 120 μM Cd^2+^ for 3 and 4 days.

**Figure 5 materials-11-02580-f005:**
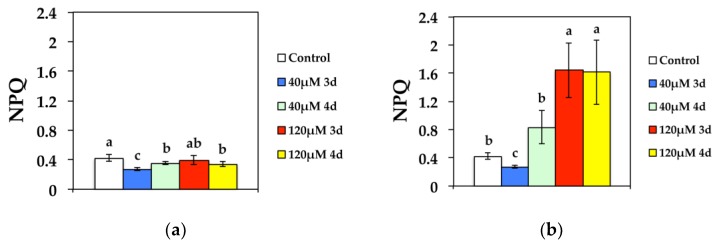
Changes in non-photochemical fluorescence quenching (NPQ) measured at (**a**) 300 μmol photons m^−2^ s^−1^ or (**b**) 1000 μmol photons m^−2^ s^−1^. *N. caerulescens* plants were grown at 0 (control), 40 or 120 μM Cd^2+^ for 3 and 4 days.

**Figure 6 materials-11-02580-f006:**
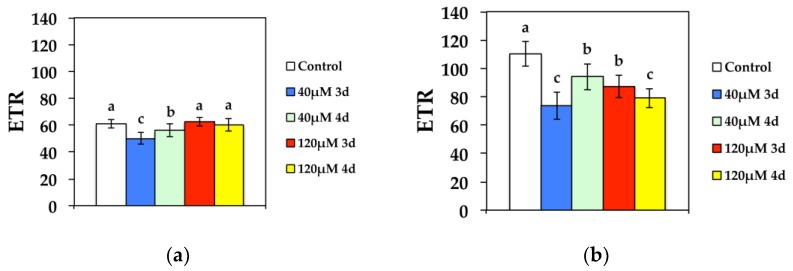
Changes in the relative PSII electron transport rate (ETR) measured at (**a**) 300 μmol photons m^−2^ s^−1^ or (**b**) 1000 μmol photons m^−2^ s^−1^. *N. caerulescens* plants were grown at 0 (control), 40 or 120 μM Cd^2+^ for 3 and 4 days.

**Figure 7 materials-11-02580-f007:**
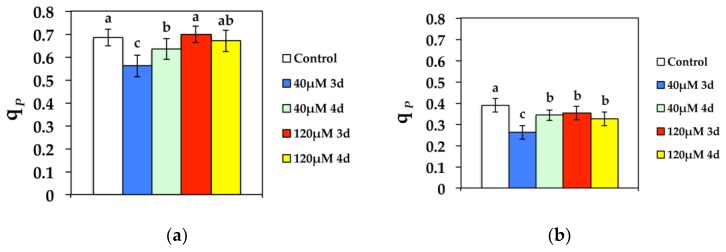
Changes in the photochemical fluorescence quenching, that is the relative reduction state of Q*_A_*, reflecting the fraction of open PSII reaction centers (*q*_P_) measured at (**a**) 300 μmol photons m^−2^ s^−1^ or (**b**) 1000 μmol photons m^−2^ s^−1^. *N. caerulescens* plants were grown at 0 (control), 40, or 120 μM Cd^2+^ for 3 and 4 days.

**Figure 8 materials-11-02580-f008:**
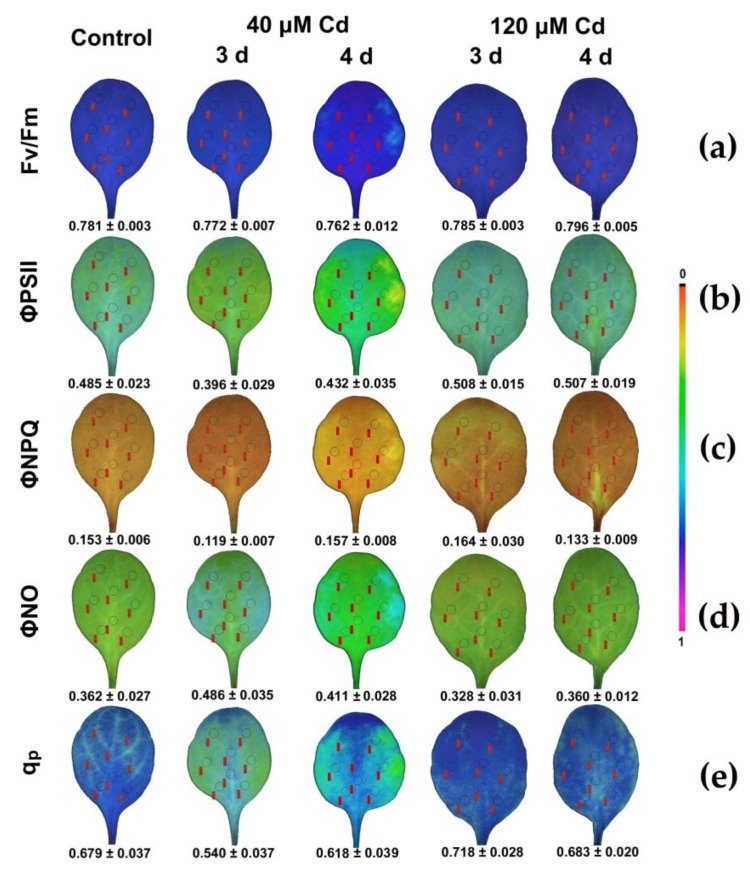
Representative chlorophyll fluorescence images of the maximum quantum efficiency (*F*_v_/*F*_m_) of PSII after 15 min dark adaptation (**a**) and after 5 min illumination at 300 μmol photons m^−2^ s^−1^ actinic light; of the actual (effective) quantum yield of PSII photochemistry (Φ*_PSII_*) (**b**), the quantum yield for dissipation by downregulation in PSII (Φ*_NPQ_*) (**c**), the quantum yield of non-regulated energy loss in PSII (Φ*_NO_*) (**d**), and the relative reduction state of Q*_A_*, reflecting the fraction of open PSII reaction centers (*q*_P_) (**e**). *N. caerulescens* plants were grown at 0 (control), 40 or 120 μM Cd^2+^ for 3 and 4 days. The colour code depicted at the right side of the images ranges from black (pixel values 0.0) to purple (1.0). The eight areas of interest are shown in each image. The average value of each photosynthetic parameter of the leaf is presented in the figure.

**Figure 9 materials-11-02580-f009:**
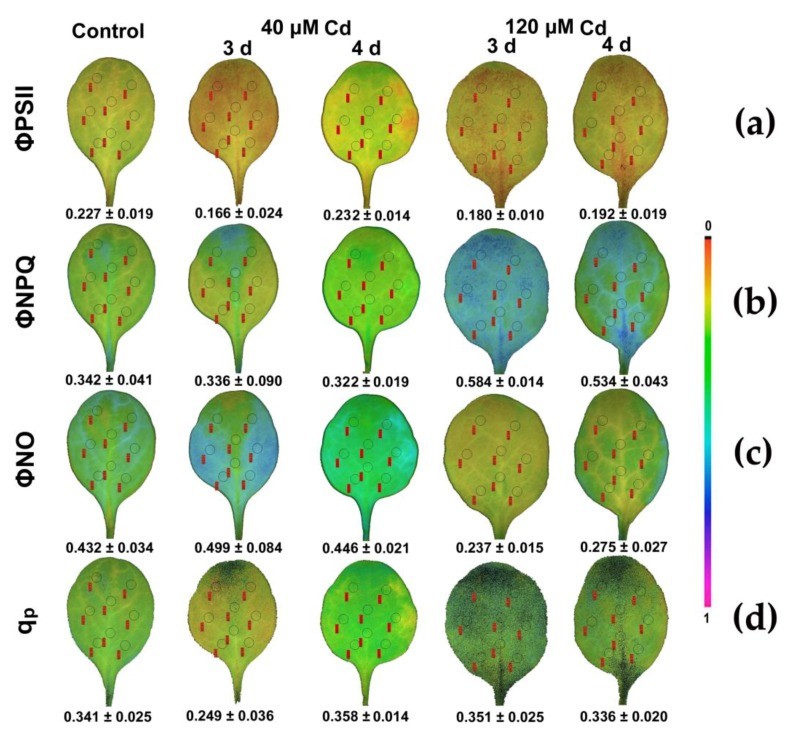
Representative chlorophyll fluorescence images after 5 min illumination at 1000 μmol photons m^−2^ s^−1^ actinic light; of the actual (effective) quantum yield of PSII photochemistry (Φ*_PSII_*) (**a**), the quantum yield for dissipation by downregulation in PSII (Φ*_NPQ_*) (**b**), the quantum yield of non-regulated energy loss in PSII (Φ*_NO_*) (**c**), and the relative reduction state of Q*_A_*, reflecting the fraction of open PSII reaction centers (*q*_P_) (**d**) *N. caerulescens* plants were grown at 0 (control), 40 or 120 μM Cd^2+^ for 3 and 4 days. The colour code depicted at the right side of the images ranges from black (pixel values 0.0) to purple (1.0). The eight areas of interest are shown in each image. The average value of each photosynthetic parameter of the leaf is presented in the figure.

**Table 1 materials-11-02580-t001:** Definitions of all measured and calculated chlorophyll fluorescence parameters.

Chlorophyll Fluo-Rescence Parameter	Definition	Calculation
*F* _o_	Minimum chlorophyll *a* fluorescence in the dark-adapted leaf (PSII centers open)	Obtained by applying measuring photon irradiance of 1.2 μmol photons m^−2^ s^−1^
*F* _m_	Maximum chlorophyll *a* fluorescence in the dark-adapted leaf (PSII centers closed)	Obtained with a saturating pulse (SP) of 6000 μmol photons m^−2^ s^−1^
*F* _s_	Steady-state photosynthesis	Measured after 5 min illumination time before switching off the actinic light (AL) of 300 μmol photons m^−2^ s^−1^ or 1000 μmol photons m^−2^ s^−1^
*F*_o_′	Minimum chlorophyll *a* fluorescence in the light-adapted leaf	It was computed by the Imaging Win software (Heinz Walz GmbH, Effeltrich, Germany) as Fo′ = Fo/(Fv/Fm + Fo/Fm′) [42]
*F*_m_′	Maximum chlorophyll *a* fluorescence in the light-adapted leaf	Measured with saturating pulses (SPs) every 20 s for 5 min after application of the actinic light (AL) of 300 μmol photons m^−2^ s^−1^ or 1000 μmol photons m^−2^ s^−1^
*F*_v_/*F*_m_	The maximum quantum efficiency of PSII photochemistry	Calculated as (*F*_m_ − *F*_o_)/*F*_m_
Φ*_PSII_*	The effective quantum yield of photochemical energy conversion in PSII	Calculated as (*F*_m_′ − *F*_s_)/*F*_m_′
*q* _P_	The redox state of QA	Calculated as (*F*_m_′ − *F*_s_)/(*F*_m_′ − *F*_o_′)
NPQ	The non-photochemical quenching that reflects heat dissipation of excitation energy	Calculated as (*F*_m_ − *F*_m_′)/*F*_m_′
ETR	The relative PSII electron transport rate	Calculated as Φ*_PSII_* x Photosynthetic Photon Flux Density × 0.5 × 0.84
Φ*_NPQ_*	The quantum yield of regulated non- photochemical energy loss in PSII, that is the quantum yield for dissipation by down regulation in PSII	Calculated as *F*_s_/*F*_m_′ − *F*_s_/*F*_m_
Φ*_NO_*	The quantum yield of non-regulated energy loss in PSII	Calculated as *F*_s_/*F*_m_
1 − *q*_P_	The fraction of closed PSII reaction centers	Calculated as 1 − *q*_P_

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
