# Peer review of "Chlorophyll Fluorescence Imaging Analysis for Elucidating the Mechanism of Photosystem II Acclimation to Cadmium Exposure in the Hyperaccumulating Plant Noccaea caerulescens"

_materials, 2018, doi:10.3390/ma11122580_

Round 1
Reviewer 1 Report
Attached file

Author Response
Some major points that need to be checked and corrected are as follows:Thank you for your comments that were very useful in improving our manuscript.The sentence in the Abstract: “This acclimation achieved through the possible Cd detoxification mechanism done by vacuolar sequestration and the reduced plastoquinone (PQ) pool that mediated stomatal closure that decreased Cd supply at the affected leaf area”, is more appropriate to move in the section conclusion, because the authors have not directly measured the reduced plastoquinone (PQ) pool and stomatal closure in this MS. They provide only indirect evidence.We agree with your comments and we moved this sentence in the section conclusion. However we maintained the phrase “This acclimation achieved possibly through the reduced plastoquinone (PQ) pool signaling” since we have estimated the redox state of QA.For the same reason, “patchy stomatal behavior” can be removed from the keywords.
Yes we do agree and wemoved it from the keywords.
I think that Table 1 given in Materials and Methods does not contain new information. The description of the chlorophyll fluorescence parameters has already been given by Rohacek, 2002 Photosynthetica, Kramer et al., 2004 Photosynth. Res., Baker, 2008 Annu. Rev. Plant Biol., etc. The authors have also published articles with this method before [Refs. 13,43,79], so they can briefly describe the parameters in the text and give citations.
Table 1 does not really contain any new information but it was included in order to decrease plagiarism from definitions of the chlorophyll fluorescence parameters that is much higher when definitions are given in the text than in a table format. Thus, table 1 was included only to reduce plagiarism.
The parameters given on the axes in Figures 2,3,4 should be changed, as
given in the text by subscript formatting (ΦPSΙΙ, ΦNPQ, ΦNO).We changed the parameters given on the axes in Figures 2,3,4 and 7 in
order to match with the subscript formatting of the text.
In my opinion, in the section “3.5. Spatiotemporal variation of PSII
responses to Cd exposure” the authors should describe only the data for
the spatiotemporal leaf heterogeneity as the changes in the parameter
values have been shown and described in the previous sections,
Figures 1-4,7.
We share this comment, thus we tried to avoid description of the changes in the parameter values that were described in the previous sections. However, it was not possible to avoid it completely.
Reviewer 2 Report
The manuscript "Chlorophyll Fluorescence Imaging Analysis for Elucidating the Mechanism of Photosystem II Acclimation to Cadmium Exposure in the Hyperaccumulating Plant Noccaea caerulescens", by Bayçu and Colleagues, presents a genuine research aimed at clarifying the mechanisms of photosystems acclimatation to high Cd level in a metallophyte plant.
The authors treated plants with two levels of cadmium for 3 and 4 days. They detected several indicators of plant adaptation, whose effects seemed to be either time-dependent, when low Cd exposure was applied, or dose-dependent, when plants were exposed to high Cd levels. Consequently, these data are finally discussed in the framework of a "Threshold for tolerance" model.
This is overall a good quality paper, suited for publication on "Materials" journal. I found, however that the description of the experimental design is not sufficiently developed, since the authors do not directly describe some crucial aspects of their experiment, but rather refer to two previous papers, one of which does not seem so strictly related to the methods used in this work. Particularly as for statistical analysis, I think the author should make the effort to clarify some questions that I pose in the following.
General concerns and suggestions:
69-73 - Experimental design - How many replicates did you use for each treatment? Is it 5, as it seems by reading at line 88?
77 - Ref n° 41 - On a first look, the quoted paper does not seem really linked to the method described. Please check the reference or describe more extensively the methods used.
88 Ref n° 43 - The quoted paper is not a statistic paper, so it is not so clear what the link between the quoted paper and Student's t-test is.
88 - Student t-test is commonly used for comparing two distributions, whereas the notations in the figures (different letters for significantly different means) seem rather to be the result of a multiple comparison. One-way Anova should be preferred in this case, in order to avoid Type I error increase.
88 - n=5 means that 5 leaves from 5 different plants were used for measuring? or the measure was repeated 5 times on the same leaves? It is important to know if the variability of the means expresses the variability among plant replicates or is just the error of the measurement.
Tab. 1 - check: "qP The the redox state of QA" (repetition of "the").
196 - check "LL" (probably GL). If it just stands for "low light", it must be written as such, since no variable abbreviation was given as LL.
Fig. 8 and 9 - AOI - although this abbreviation was defined in the methods (line 78), it would be fine for the reader if it is expressly explained also in the figure legends.
238 - Was Hydrogen peroxide level measured in this experiment, or is this just an interpretation based on analogy with paper 61? If so, it would be better to express it as "probably through the generation ..." (line 240), or any other formulation to distinguish data from mechanism interpretation.
266-267 - ...exhibit spatially and temporally patterns. Not so clear, could you please rephrase?
274 - ...caused significantly effects (it seems rather "significant effects").
Author Response
This is overall a good quality paper, suited for publication on "Materials" journal. I found, however that
the description of the experimental design is not sufficiently developed, since the authors do not
directly describe some crucial aspects of their experiment, but rather refer to two previous papers, one of
which does not seem so strictly related to the methods used in this work. Particularly as for statistical analysis,
I think the author should make the effort to clarify some questions that I pose in the following.
Thank you for your comments that were very useful in improving our manuscript.
General concerns and suggestions:
69-73 - Experimental design - How many replicates did you use for each treatment? Is it 5, as it seems
by reading at line 88?
We have omitted the extensive description of the experimental design that was included in
our first version of our manuscript in order to decrease plagiarism. We used 15 plants per treatment from
which 5 leaves were measured from 5 different plants. We inserted this information on lines 77-78 in
our revised manuscript
77 - Ref n° 41 - On a first look, the quoted paper does not seem really linked to the method described.
Please check the reference or describe more extensively the methods used.
Yes, from a first look as you say the paper does not seem really linked to the method described.
As we mentioned previously we omitted the extensive description of the method due to plagiarism.
In the revised manuscript we cite two references of our previous works that describe in detail the method
we used; one reference that is a similar work and the other with a more extensive description but on a first
look not seem really linked to the method described. Plagiarism check that is positive to my opinion, results
in some negative effects when it has to do with the description of previously used methodology.
88 Ref n° 43 - The quoted paper is not a statistic paper, so it is not so clear what the link between the
quoted paper and Student's t-test is.
We omitted the quoted paper since the only reason that it was included was the similarity of this
sentence to that in the quoted paper.
88 - Student t-test is commonly used for comparing two distributions, whereas the notations in the figures (different letters for significantly different means) seem rather to be the result of a multiple comparison. One-way Anova should be preferred in this case, in order to avoid Type I error increase.
We will have that in mind in our featured statistical analyses.
88 - n=5 means that 5 leaves from 5 different plants were used for measuring? or the measure was repeated
5 times on the same leaves? It is important to know if the variability of the means expresses the
variability among plant replicates or is just the error of the measurement.
We elucidate this in lines 87-88.
Tab. 1 - check: "qP The the redox state of QA" (repetition of "the").
Thank you for pointing it.
196 - check "LL" (probably GL). If it just stands for "low light", it must be written as such, since no
variable abbreviation was given as LL.
Thank you also for pointing this, it is GL. We have used at the beginning the symbols LL for
"low light" and HL for "high light" but since the LL was the growth light we changed it to GL.
Fig. 8 and 9 - AOI - although this abbreviation was defined in the methods (line 78), it would be fine
for the reader if it is expressly explained also in the figure legends.
I do share this opinion. Thus we omitted the abbreviation "AOI" and we used instead "areas of interest" through out the manuscript (it is mentioned only 3 times).
238 - Was Hydrogen peroxide level measured in this experiment, or is this just an interpretation based on analogy with paper 61? If so, it would be better to express it as "probably through the generation ..." (line 240), or any other formulation to distinguish data from mechanism interpretation.
Yes, you are right. We did not measured hydrogen peroxide level and thus we changed the sentence as you suggested.
266-267 - ...exhibit spatially and temporally patterns. Not so clear, could you please rephrase?
We changed it to spatiotemporal patterns. This is the phrase the authors of the article use.
274 - ...caused significantly effects (it seems rather "significant effects").
We changed it to “caused significant effects on PSII functioning”.
Reviewer 3 Report
In the manuscript titled with “The Chlorophyll Fluorescence Imaging Analysis for Elucidating the Mechanism of Photosystem II Acclimation to Cadmium Exposure in the Hyperaccumulating Plant Noccaea caerulescens”, the authors studied photosynthetic acclimation to Cd toxicity using the hyperaccumulator Noccaea caerulescens as a model system. Photosynthesis has been shown to be very sensitive to Cd and the authors characterized the of chlorophyll fluorescence parameters acclimation after Cd exposure. The authors showed that Noccaea caerulescens leaves grown under different growth conditions show a spatial heterogeneity in PSII functionality. And they proposed that the response of Noccaea caerulescens to different Cd exposure fits the ‘Threshold for Tolerance Model’.
Overall the manuscript is well written, and it fits the scope of the journal. The references also reflect the recent progress in the field. The following are my major comments:
1. In Fig 1-7, the labels on the bar graphs are misleading. Instead of repeating the same sentence in each figure: ‘Columns with same letter are not statistically different (P < 0.05)’, the authors should consider either remove them or clarify in the legend of the figures or the main text.
2. In Fig 8 and Fig 9, the authors should add scale bar to the whole figure. It seems that the size of the leaves is different for the image of control, 3 day and 4 day treatment with 40 uM Cd, however this does not hold for the data set with 120 uM Cd treatment. The authors should give an explanation why the size of the leaves is lag time and Cd concentration dependent (if the trend is real).
3. Regarding the ‘Threshold for Tolerance Model’ (in the conclusion section), the authors should give more details about the model selection including the reference if possible. What is the basic assumption of this model? Why the authors choose this model to interpret the data? The authors should consider to further discuss the biological significance of the lag time and Cd concentration in this model.
4 For the growth light(GL) and high light( HL) comparison, the authors should justify the choice of the intensity range.
5 The authors should consider rewrite the abstract to make it more accessible to general audience.
Author Response
Overall the manuscript is well written, and it fits the scope of the journal. The references also reflect the recent progress in the field. The following are my major comments:
Thank you for your comments that were very useful in improving our manuscript.
1. In Fig 1-7, the labels on the bar graphs are misleading. Instead of repeating the same sentence in each figure: ‘Columns with same letter are not statistically different (P < 0.05)’, the authors should consider either remove them or clarify in the legend of the figures or the main text.
We omitted this sentence from figures 1-7 with incorporation of the relevant information on Statistical analyses section.
2. In Fig 8 and Fig 9, the authors should add scale bar to the whole figure. It seems that the size of the leaves is different for the image of control, 3 day and 4 day treatment with 40 uM Cd, however this does not hold for the data set with 120 uM Cd treatment. The authors should give an explanation why the size of the leaves is lag time and Cd concentration dependent (if the trend is real).
The scale bar is the same in all leaves. We substitute the representative leaf of 4 d exposure to 40 μM Cd with another one. Thus, there is not any difference in size from the others. We have noticed a lag time size of the leaves after 10 d exposure to Cd (in our previous work).
3. Regarding the ‘Threshold for Tolerance Model’ (in the conclusion section), the authors should give more details about the model selection including the reference if possible. What is the basic assumption of this model? Why the authors choose this model to interpret the data? The authors should consider to further discuss the biological significance of the lag time and Cd concentration in this model.
Thank you for pointing this. We apologize for our mistake not to give more details about the model and also any references. We have included more details regarding the ‘Threshold for Tolerance Model’ (lines 267-273).
4 For the growth light (GL) and high light (HL) comparison, the authors should justify the choice of the intensity range.
A sentence has been inserted for the choice of the intensity range (lines 78-81).
5 The authors should consider rewrite the abstract to make it more accessible to general audience.
We rewrote the abstract hopping is more accessible to a wider audience. However since the topic is specialized we don’t believe we have done a good job.